# Protective Efficacy of Subunit Vaccine Expressing Rv0976c Against Tuberculosis

**DOI:** 10.3390/vaccines13080872

**Published:** 2025-08-17

**Authors:** Ziwei Zhou, Dan Chen, Fuzeng Chen, Wenxi Xu, Zhifen Pan, Zhihao Xiang, Xiaoxiao Gao, Yeyu Li, Fagang Zhong, Jun Liu, Lu Zhang

**Affiliations:** 1State Key Laboratory of Genetic Engineering, Institute of Genetics, School of Life Science, Fudan University, Shanghai 200433, China; zhou_ideal@163.com (Z.Z.); fuzeng910@163.com (F.C.); xiangzhh91@163.com (Z.X.); gaoxiaoxtx@163.com (X.G.); 18826072476@163.com (Y.L.); 2Department of Microbiology, School of Life Science, Fudan University, Shanghai 200433, China; chendan0623@hotmail.com; 3Princess Margaret Cancer Center, Toronto, ON M5S 1A8, Canada; wenxi.xu2@uhn.ca; 4Department of Tuberculosis, The First Hospital of Jiaxing, Jiaxing University, Jiaxing 314001, China; pzf_zj@yeah.net; 5Xinjiang Academy of Agricultural and Reclamation Science, Shihezi 832000, China; zfg125@sohu.com; 6Department of Molecular Genetics, University of Toronto, Toronto, ON M5S 1A8, Canada; jun.liu@utoronto.ca; 7Shanghai Engineering Research Center of Industrial Microorganisms, Shanghai 200433, China

**Keywords:** tuberculosis, antigens, subunit vaccines, Rv0976c, protective efficacy

## Abstract

Objectives: The construction of subunit vaccines based on antigens that can induce strong cellular immunity is a widely accepted strategy to develop new tuberculosis vaccines. This study screens immunogens with potential for subunit vaccine development from seven candidate antigens and then verifies their vaccine efficacy. Design: C57BL/6 mice were immunized subcutaneously with purified PPE19, PPE50, FadD21, Rv1505c, Rv1506c, Rv2035, and Rv0976c proteins formulated with Freund’s adjuvant to evaluate both the antigen-specific Th1 cellular immune responses and IgG level. After the vaccination of mice with recombined pcDNA3.1 expressing Rv0976c, intravenous or aerosol infection with *M. tb* were further challenged to assess protective efficacy. Results: Purified PPE19, PPE50, FadD21, and Rv0976c proteins generated strong antigen-specific Th1 cellular immune responses in mice. Compared to Ag85A, Rv0976c also stimulated higher IgG antibody level in mice. In particular, Rv0976c stimulated high and specific IgG antibody levels in serum from TB patients. The vaccination of mice with DNA vaccines expressing Rv0976c, followed by intravenous challenge with Bacillus Calmette–Guerin (BCG) Pasteur or *M. tb*, resulted in significant levels of protection that are comparable to or better than that afforded by the two leading antigens, Ag85A and PPE18. Conclusions: These results indicated that Rv0976c was a better protective antigen. Future studies to combine Rv0976c with other antigens and evaluate its effectiveness as a booster of BCG or as a therapeutic vaccine are warranted.

## 1. Introduction

Tuberculosis (TB), caused by the bacillus *Mycobacterium tuberculosis* (*M. tb*), remains one of the top 10 causes of death worldwide, with 10.8 million people having developed TB and 1.25 million having died as a result in 2023 [1]. The emergence of drug-resistant TB and co-infection with HIV and COVID-19 have made the prevention and therapy of TB more difficult. A vaccine for the prevention of tuberculosis is the first and the most important line of defense against TB. *Bacillus-Calmette–Guerin* (BCG), the only admitted TB vaccine, is an attenuated strain, derived from *Mycobacterium bovis* (*M. bovis*), with many virulence factors lost due to being subcultured 230 times in vitro. Although BCG could provide about 80% protection to TB in infants, in adolescents and adults, its protection rate was lower and fluctuated greatly [2], which may be caused by the variability in environment [3].

To improve the efficiency of BCG, the heterologous prime–boost strategy should be used to enhance the immune response introduced by the BCG vaccine in the next generation of TB vaccines [4]. The strategy includes vaccinating infants and young children with BCG or recombinant BCG prior to TB exposure, followed by a booster immunization with a subunit vaccine (mainly is protein, peptide, or DNA). The subunit vaccine may also be independently administered to adolescents or utilized as adjunctive immunotherapy alongside standard chemotherapy regimens. Subunit vaccines, which focused on expressing immunodominant antigens to elicit an *M. tb*-specific immunological response, first emerged in the early 1980s and achieved a great breakthrough in the 1990s.

The screening of antigens is the most important part of subunit vaccine research and development. Besides stimulating CD8^+^ T cells and Th17 cells, BCG mainly stimulates CD4^+^ T cells to induce Th1 immune responses, and the expression level of Th1 cytokines including IL-2, TNF, and IFN-γ are up regulated [5,6]. As an important cytokine in host defense against *M. tb* infection [7], the expression level of IFN-γ was regarded as a criterion to evaluate the immune response induced by BCG and subunit vaccines [8]. Antigens which could stimulate CD4^+^ T cell immune response to produce a high level of IFN-γ that have currently been identified include the Esx family proteins (EsxA, B, G, H, G, N), the antigen 85 family (Ag85A, B, C), and several PE/PPE family proteins (e.g., PPE18, PPE14) [9,10,11]. In addition, recognizing that the antigenic repertoire of *M*. *tb* changes with different infection stages, multi-stage subunit vaccines with fused antigens which drive T cell functional heterogeneity have received great attention. Many of them have now achieved encouraging results in the early stage of clinical trials. Three fusion proteins, including H56 (Rv1886c, Rv3875, Rv2660c) [12], ID93 (Rv1813, Rv2603, Rv3619, Rv3620) [13], and M72 (PPE18, Rv0125) [14], have been constructed and entered phase II phase III clinical trials [1] and all these three fusion proteins were shown to induce a high level of antigen-specific IFN-γ in animal experiments. In addition, to better stimulate CD8^+^ T cells, subunit vaccines have been constructed by combining candidate antigens or DNA, including MVA85A, a vaccinia virus expressing Ag85A, and AERAS-402, an adenovirus-35 expressing Ag85A, Ag85B, and TB10.4 [15].

As part of our effort to identify new antigens for vaccine development, we have focused on several classes of proteins, including the latency-associated proteins Rv0976c, Rv1505c, Rv1506c, and Rv2035, PE/PPE family proteins PPE19 and PPE50, and enzymes involved in lipid metabolism FadD2, which we had discovered in dormancy and reactivation of transcriptomics of *M. tb* strain. Though the functional category of Rv0976c, Rv1505c, Rv1506c, and Rv2035 are hypothetical proteins, our previous research has demonstrated that the Rv1501-1508c fragment of the M. tb genome is associated with virulence, BCG::*Rv1501-1508c* showed an enhanced anti-tuberculosis protective effect [16]; meanwhile, Rv2035 belongs to the toxin–antitoxin system (TA system) and may participate in lipid metabolism regulation [17,18]. Here, we constructed the recombinant plasmids, obtained the recombinant proteins, and described the immunogenicity and protective efficacy of these proteins. Our results indicate that Rv0976c is a promising antigen that could be included in novel subunit vaccine construction.

## 2. Materials and Methods

### 2.1. Bacterial Strains and Development Conditions

*Mycobacterium tuberculosis* H37Rv was grown at 37 °C in Middlebrook 7H9 broth (Difco™, Lot No: 271310), which was supplemented with 0.2% glycerol, 10% albumin–dextrose–catalase (ADC; BD BBL™, NJ, USA, Lot No: 211877) and 0.05% Tween 80. Middlebrook 7H10 agar (Difco™, BD, NJ, USA, Lot No: 262710) was used for solid culture, which was supplemented with 0.5% glycerol and 10% oleic acid–albumin–dextrose–catalase (OADC; BD BBL™, NJ, USA, Lot No: 212351).

*Escherichia coli* DH5α and BL21was developed at 37 °C in LB broth or agar, which was used for the propagation of plasmid DNA and expression and purification of recombinant proteins, respectively.

### 2.2. Molecular Cloning

For expression in *E. coli*, the ORF of each selected gene was amplified by PCR using the appropriate primers listed in Appendix A. The PCR products were digested with the corresponding enzymes (Appendix A) for 3 h at 37 °C. Plasmid pET28a (Novagen, CA, USA) or pET SUMO (Invitrogen, CA, USA) were digested with the same enzymes, and these two fragments were ligated, transformed into *E. coli* DH5α, and plated on LB agar containing kanamycin (50 µg/mL). After overnight incubation at 37 °C, single colonies were randomly picked and grown in LB broth. The plasmids were isolated from *E. coli* DH5α culture using an Axygen Miniprep Kit (CA, USA) and confirmed by DNA sequencing.

For expression in mammalian cells, the ORF of selected genes was cloned into the eukaryotic expression vector pVAX1 or pcDNA3.1 (+) (Life Technologies, NY, USA). The PCR primers for cloning are listed in Appendix A. Each forward primer used in PCR reactions contains a Kozak sequence and the reverse primer contains a His tag sequence at the C-terminal end. The final constructs were sequenced to verify the accuracy of the gene sequence and purified using an EndoFree Maxi Plasmid Kit (TIANGENgERMANY). The expression of each construct in mammalian cells was confirmed by transfecting 293T or HeLa cells (ATCC) using the lipofectamine LTX kit (Life technologies, NY, USA), Lot No: 13778-075), followed by Western blot analysis using an antibody against His tag.

### 2.3. Expression and Purification of Recombinant Proteins

The pET28 and pET SUMO constructs that had been enriched from *E. coli* DH5α were transformed into *E. coli* BL21, respectively, and the recombinant *E. coli* BL21 was cultured on the LB plate containing 50 µg/mL kanamycin overnight to gained single colonies. As for the soluble fraction of target proteins, the collected supernatant was co-incubated with Ni-NTA His•Bind^®^ Resin (Novagen, Lot No: 70666-3) overnight and purified according to the manufacturer’s protocol. As for the insoluble fraction of target proteins, after being sonicated and centrifuged, the precipitation was resuspended with the mixture containing 20 mM Tris (pH 8.0), 100 mM NaCl, and 8 M urea, and then co-incubated with Ni-NTA His•Bind^®^ Resin overnight and purified. Finally, at 4 °C, the purified proteins were centrifuged at 4000 rpm until the urea and imidazole were cleared. SDS-PAGE (Genescript) was used to make sure the protein purity. The endotoxin of proteins was removed with the ToxinEraser Endotoxin Removal Kit (Genscript, CA, USA) and the endotoxin level of protein was measured by the same kit. The protein concentration was measured by the BCA assay (Sigma, MO, USA, Lot No: B9643).

### 2.4. Immunogenicity Assays

Animal-related experiments were conducted under the approval of the animal care committees. After being mixed with 100 µL of Freund’s incomplete adjuvant (Sigma, MO, USA, Lot No: F5581), 100 µL of every purified protein containing 10 µg was injected subcutaneously into four C57BL/6 mice. At 2 weeks and 4 weeks, these mice were immunized again. On the 8th week, they were sacrificed and the spleens and whole blood of each group were collected.

The spleens were gently ground to separate the splenocytes, which were further developed in a 24-well plate with an amount of 10^6^ per well. To enhance immune level and activate antigen-specific memory of T cells to improve the corresponding spleen-specific immune functions, which are conducive to the detection of FACS or ELISA, the splenocytes were stimulated by purified proteins of 5 and 10 µg/mL and developed at 37 °C with 5% CO_2_ for 72 h. The cell supernatants were collected and centrifuged at 4000 rpm for 10 min at 4 °C; then, the concentration of Th1 cytokines, including IFN-γ, TNF, and IL-2, were determined by ELISA (OptEIATM ELISA Kit of BD Biosciences, NJ, USA) according to the recommended procedure.

The whole blood was collected in aseptic test tubes without anticoagulant and then centrifuged at 2400 rpm for 10 min after being placed at room temperature for 4 h. The serum was collected in a cryopreservation tube and stored at −80 °C. The levels of IgG, IgG1, and IgG2c antibodies in the serum of the mice were determined by ELISA, using the OptEIATM ELISA Kit (BD Biosciences, NJ, USA), with appropriate mAbs, according to the protocol recommended by the manufacturer.

### 2.5. Evaluation of Serum Antibody Levels of Rv0976c 

In this experiment, a total of 50 serum samples were collected, which were stored in a refrigerator at −80 °C. All serum samples from the first Hospital of Jiaxing were HIV-negative. The serum samples were divided into two groups: (1) 30 patients infected with *M. tb*, including 13 of active TB and 17 of latent infections; (2) 20 healthy controls who had been vaccinated with the *M. bovis* BCG vaccine.

Rv0976c, Ag85A, and ESAT-6 antigens were coated onto Elisa plates, respectively, and the serum collected from TB patients and healthy controls were diluted at 1:100 as primary antibodies to detect whether the protein had a high antibody response in human serum, according to the protocol recommended above.

### 2.6. Animal Protection Assays

For intravenous infection experiments, BALB/c mice (6 per group) were immunized subcutaneously three times (2 weeks apart) with 100 μg DNA constructs, and 8 weeks after the first vaccination, the mice were challenged intravenously with 1.49 × 10^7^ CFUs of BCG Pasteur or 10^5^ CFUs of *M. tb* H37Rv. Three weeks after BCG Pasteur challenge or four weeks after *M. tb* challenge, the mice were sacrificed and the presence of *bacteria* colony-forming units (CFUs) in the lungs and spleens was determined by serial dilution and plating on 7H10 agar plates (OADC). For both experiments, at day one post-infection, one mouse of each group was sacrificed and the lungs were harvested, homogenized in PBS, and plated on 7H10 agar to enumerate the bacterial burden. This was performed to confirm the actual infection dosage.

## 3. Results

### 3.1. Cellular and Humoral Immune Responses Induced by Selected Proteins

*Lsr2* is a global transcriptional regulator of *M. tb* and has been implicated in multiple cellular processes, including the establishment of persistent infection [19,20,21,22,23,24]. In our unpublished study, we found that deletion of *lsr2* in *M. tb* led to significant increases in the expression of four genes: *Rv1506c* (24.9-fold)*, Rv1505c* (17.6-fold)*, Rv0976c* (15.3-fold), and *Rv2035* (11.5-fold), compared to the wild-type strain. These findings suggest that proteins encoded by these genes may be associated with the latent stage of infection. All four are conserved hypothetical proteins and have not been tested in vaccine development. Accordingly, these proteins were selected for further evaluation of their immunogenicity and potential to serve as protective antigens in TB vaccine development. 

For comparison, we included Ag85A and PPE18, two antigens with known protective effects, in our study. Ag85A is an enzyme involved in lipid metabolism [25] and PPE18 is a representative member of the PE/PPE protein family [26]. We also included three additional proteins based on their biological relevance: FadD21 (*Rv1185c*), an enzyme involved in lipid metabolism, and two PE/PPE family proteins, PPE19 (Rv1361c) and PPE50 (Rv3135). FadD21 is essential for *M. tb* survival in the spleen of C57BL/6J mice [27]. PPE19 shares a high sequence similarity to PPE18 (84.9% identity) [28] and PPE50 was the only PE/PPE gene found to be upregulated in both BCG-Japan and BCG-Pasteur compared to M. tb and *M. bovis* [28,29]. To date, none of these proteins have been evaluated as vaccine candidates.

To determine the immunogenicity of selected proteins, each open reading frame (ORF) was cloned into pET28a vector, expressed in *E. coli* BL21, and purified (Figure 1). Recombinant Rv0976c, Rv1505c, Rv1506c, Rv2035, and Ag85A proteins were purified from the soluble fraction using the standard procedures. In contrast, FadD21, PPE18, PPE19, and PPE50 were isolated from the insoluble fraction (inclusion body) under denaturing conditions followed by renaturation.

To assess the immunogenicity of target antigens, C57BL/6 mic were divided into groups, with four in each, and immunized three times (two weeks apart) with 10 μg endotoxin-free protein formulated with Freund’s complete adjuvant or Freund’s incomplete adjuvant. Eight weeks after the first vaccination, the mice were sacrificed and their spleens were all collected. Then splenocytes were prepared by grinding, and after cultivating and restoring the vitality of the cells, the splenocytes were stimulated with or without the corresponding protein at two different concentrations (5 and 10 μg/mL). Three days after stimulation with antigens, the cell supernatants were harvested and the production of Th1 cytokines (IFN-γ, TNF, IL-2) was determined by ELISA (Figure 2). 

The three PE/PPE proteins, PPE18, PPE19, and PPE50, induced the highest level of IFN-γ (~600 pg/mL). PPE19 induced equally high levels of TNF but much lower levels of IL-2. Compared to PPE19, PPE18 and PPE50 induced lower levels of TNF and higher levels of IL-2 and the three cytokines’ (IFN-γ, TNF, IL-2) levels of FadD21 were comparable to PPE18. Ag85A induced higher levels of IFN-γ than TNF or IL-2 (IFN-γ:TNF:IL-2 = 1.7:1.2:1.0). Of all proteins tested, Rv0976c had the most balanced ratio of the three cytokines, with nearly equal amounts of IFN-γ and IL-2 and ~50% more TNF (IFN-γ:TNF:IL-2 = 1.4:1.8:1.0). The other three conserved hypothetical proteins, Rv1505c, Rv1506c, and Rv2035, induced very low levels of TNF (<80 pg/mL) and non-detectable levels of IFN-γ and IL-2. 

Before spleen collection, whole blood was collected, and the serum was isolated. The serum levels of IgG, IgG1, and IgG2c were measured by ELISA to evaluate the humoral immunity responses induced by selected proteins (Figure 3). Among all tested proteins, both Rv0976c and Ag85A stimulated strong antibody responses, with IgG titers beginning to decline beyond a 1:6400 dilution. The decrease was more pronounced in the Rv0976c group. All of the tested proteins stimulated detectable antibody responses, with the Ag85a showing the highest overall titers. Additionally, under the same assay conditions, Rv0976c induced high IgG levels in serum from TB patients, demonstrating comparable specificity to ESAT-6 and greater specificity than Ag85A (Appendix A).

Taken together, the cellular and humoral immune responses elicited by Rv0976c, FadD21, PPE19, and PPE50 suggest their potential as promising vaccine candidates. Therefore, these four antigens were selected for further evaluation in protection experiments.

### 3.2. Protective Efficacy of Selected Antigens

We elected to test the protective efficacy of the above antigens in the form of DNA vaccines since it has the potential to induce CD8^+^ T cell response. The genes of the selected antigens were cloned into the mammalian expression vector pcDNA3.1 (+) or pVAX1. Ag85A and PPE18 were included as controls and for comparison. Mammalian cells 293T or HeLa were transfected with individual constructs, and the expression of the corresponding antigen was confirmed by Western blot analysis (Appendix A).

To preliminarily determine the protective efficacy of Rv0976c, we chose the Rv0976c cloned into pcDNA3.1(+) as the performed vaccination and immunized mice followed by an intravenous challenge with BCG Pasteur. The pcDNA: Ag85A construct was included in this experiment for comparison. BALB/c mice were vaccinated with the pcDNA constructs (100μg, three times, two weeks apart). Eight weeks after the first vaccination, mice were intravenously challenged with 1.49 × 10^7^ CFUs of BCG Pasteur through the tail vein. Mice were euthanized at week 3 post-challenge to harvest the lungs and spleen, and the CFUs of BCG Pasteur in these organs were determined. As a result, mice vaccinated with pcDNA: Rv0976c exhibited significantly lower M. tb burden in the lungs (*p* < 0.0001, unpaired Student’s t-test) and spleens (*p* < 0.001, unpaired Student’s t-test) than those vaccinated with the pcDNA control (Figure 4). Mice vaccinated with pcDNA: Ag85A appeared to have lower *M. tb* burden than the pcDNA: Rv0976c group, especially in spleens (*p* < 0.05, unpaired Student’s t-test), but the difference were not statistically significant in lungs. Rv0976c was preliminarily confirmed to have a good protective effect.

To determine the protective efficacy of Rv0976c compared to different antigens, those DNA constructs were used to immunize mice, followed by intravenous challenge with *M. tb* H37Rv. BALB/c mice (six per group) were immunized subcutaneously three times (2 weeks apart) with 100 μg DNA vaccines. Eight weeks after the first vaccination, the mice were challenged intravenously with 10^5^ CFUs of *M. tb* H37Rv. Four weeks after *M. tb* challenge, the mice were sacrificed and the presence of *M. tb* colony-forming units (CFUs) in the lungs and spleen was determined. Results showed that mice immunized with pVAX: Rv0976c had significantly lower *M. tb* burden in the lungs and spleen than those vaccinated with the empty vector (pVAX) control (*p* < 0.05, unpaired Student’s t-test) (Figure 5). The average *M. tb* burdens in the pVAX: Rv0976c group were 6.28 log_10_ and 5.44 log_10_ CFUs in the lungs and spleen, respectively, which were about 0.4–0.5 log_10_ lower than in the pVAX control group (6.75 log_10_ and 5.82 log_10_ CFUs in the lungs and spleen, respectively). However, no significant differences were found between the *M. tb* burden in mice vaccinated with pVAX: Ag85A compared to the pVAX control group. For genes cloned into the pcDNA vectors, mice immunized with pcDNA: PPE18 and pcDNA: Ag85A had, on average, 6.24 log_10_ and 6.38 log_10_ CFUs of *M. tb* in the lungs, respectively, which were 0.2–0.4 log_10_ lower than mice vaccinated with the pcDNA empty vector (6.59 log_10_ CFUs), but the differences were not statistically significant. However, when compared with the pVAX control group, pcDNA: PPE18 afforded significantly better protection (*p* < 0.05, unpaired Student’s t-test), which was comparable to pVAX: Rv0976c. A similar trend was found for the spleen *M. tb* burden in mice vaccinated with these two DNA constructs. The pcDNA constructs expressing the other three antigens, FadD21, PPE19, and PPE50 did not exhibit any protective efficacy. 

To determine the protective efficacy of Rv0976c compared to different antigens, those DNA constructs were used to immunize mice, followed by intravenous challenge with *M. tb* H37Rv. BALB/c mice (six per group) were immunized subcutaneously three times (2 weeks apart) with 100 μg DNA vaccines. Eight weeks after the first vaccination, the mice were challenged intravenously with 10^5^ CFUs of *M. tb* H37Rv. Four weeks after *M. tb* challenge, the mice were sacrificed and the presence of *M. tb* colony-forming units (CFUs) in the lungs and spleen was determined. Results showed that mice immunized with pVAX: Rv0976c had significantly lower *M. tb* burden in the lungs and spleen than those vaccinated with the empty vector (pVAX) control (*p* < 0.05, unpaired Student’s t-test) (Figure 5). The average *M. tb* burdens in the pVAX: Rv0976c group were 6.28 log_10_ and 5.44 log_10_ CFUs in the lungs and spleen, respectively, which were about 0.4–0.5 log_10_ lower than in the pVAX control group (6.75 log_10_ and 5.82 log_10_ CFUs in the lungs and spleen, respectively). However, no significant differences were found between the *M. tb* burden in mice vaccinated with pVAX: Ag85A compared to the pVAX control group. For genes cloned into the pcDNA vectors, mice immunized with pcDNA: PPE18 and pcDNA: Ag85A had, on average, 6.24 log_10_ and 6.38 log_10_ CFUs of *M. tb* in the lungs, respectively, which were 0.2–0.4 log_10_ lower than mice vaccinated with the pcDNA empty vector (6.59 log_10_ CFUs), but the differences were not statistically significant. However, when compared with the pVAX control group, pcDNA: PPE18 afforded significantly better protection (*p* < 0.05, unpaired Student’s t-test), which was comparable to pVAX: Rv0976c. A similar trend was found for the spleen *M. tb* burden in mice vaccinated with these two DNA constructs. The pcDNA constructs expressing the other three antigens, FadD21, PPE19, and PPE50 did not exhibit any protective efficacy. 

## 4. Discussion

TB vaccine research has gained momentum in the past decade, with a dozen candidates currently under clinical trial evaluations [30]. However, whether the top-ranked candidate drugs can pass phase III clinical trials and registration remains uncertain. As previously mentioned, although MVA85A was the first candidate vaccine to pass the effectiveness test in clinical trials, it failed to provide a good protective effect for infants who had received BCG vaccination [19]. Several explanations for this have been proposed [30,31], including the modest magnitude of immune responses induced by MVA85A in children [10] and that the Ag85 family antigens were downregulated to very low levels soon after *M. tb* infection and not preferentially recognized by the immune system [32,33]. Due to hosts conditions during infection, *M. tb* changes its gene expression profile, resulting in a different or modified antigenic repertoire and it is not surprising that the Ag85 family genes, which encode enzymes involved in cell wall lipid biosynthesis, were downregulated during latent infection [32,33]. Given the failure of MVA85A, recent efforts to identify new antigens have been focusing on latency-associated antigens. Several latency-associated antigens have been tested as components of a subunit vaccine, including Rv2660c, Rv2031c (HspX), and Rv1813 [12,33,34,35]. Rv2660c was found to be expressed at the same level at the early (3 weeks) and late stage (20 weeks) of infection in the lungs of mice and was able to induce IFN-γ in latent *M. tb*-infected individuals, hence being considered as a latency-associated protein [32]. Rv2660c itself was of low immunogenicity and did not show any protection in mice when used alone [32]. However, when combined with two early-stage antigens, Ag85B and EsxA, the fusion protein H56 (Ag85B-EsxA-Rv2660c) conferred better protection than H1 (Ag85B-EsxA) in mice at the late-stage of *M. tb* infection [32]. Rv1813 was combined with PPE42-EsxV-EsxW, and the resulting fusion protein, ID93, was found to have a good boosting effect of BCG in guinea pigs [13]. Both H56 and ID93 have entered phase II clinical trials. Furthermore, when combined with other antigens, like EsxS, PPE44, and EsxV, Rv2031c was able to induce stronger cellular immunity in animal models [36,37]. These recent studies suggested that subunit vaccines combining multistage antigens may be a better strategy for developing effective vaccines.

The aim of our study is to identify new antigens that could be included in multistage vaccine constructions. We have selected three groups of proteins based on their biological functions and their potential roles in host infection for our analyses, including the Lsr2 regulated proteins (Rv0976c, Rv1505c, Rv1506c, and Rv2035), a protein involved in lipid metabolism (FadD21), and PE/PPE family proteins (PPE19 and PPE50). We first evaluated their ability to induce a Th1 cellular immune response and humoral immune response, then selected the positive ones for further efficacy testing. This was based on the well-established notion that although alone it does not predict protection, the Th1 response is critically required in adaptive immunity against *M. tb* infection [38]. The two PE/PPE proteins and one FadD enzyme were all immunogenic and capable of inducing significant levels of Th1 cytokines. However, consistent with the current notion that there are no immunological markers of efficacy [8], the levels of Th1 cytokines induced by these antigens did not correlate with their protection efficacy, which highlights the challenge facing the TB vaccine research community. 

The most significant finding of our study is the identification of Rv0976c as a protective antigen, as Rv0976c was comparable to PPE18 and consistently exhibited a better protection than Ag85A. This was demonstrated by BCG Pasteur and *M. tb* infection, and by cloning Rv0976c and Ag85A into two different expression vectors (pVAX1 and pcDNA3.1). In the BCG Pasteur infection model, the level of protection afforded by Rv0976c was somewhat higher than Ag85a in the mice spleens but the difference disappeared in lungs. In the *M. tb* intravenous infection model, mice immunized with Rv0976c had 0.4–0.5 log_10_ fewer *M. tb* than in the non-vaccinated mice and the difference is statistically significant. This level of protection afforded by Rv0976c is likely biologically significant because BCG typically reduces the *M. tb* burden by 0.5–1.0 log_10_ in similar experiments [35]. 

Rv0976c is a conserved hypothetical protein of *M. tb* and its homolog is present in multiple mycobacterial species including the *M. tb* clinical strain CDC1551, *M. bovis*, *M. ulcerans*, *M. paratuberculosis*, and *M. marinum*. Although its role in *M. tb* infection remains unknown, Rv0976c is likely a latency-associated antigen. This is because Rv0976c is part of the in vivo-expressed genomic island (iVEGI) that includes Rv0960-Rv1001 and occupies a contiguous region of 34.1 kb of the *M. tb* genome sequence [39]. Multiple genes of the iVEGI, including Rv0976c, were highly expressed in *M. tb* during infection of mice (BALB/c and C3HeB/FeJ mice), but they were expressed at low levels in *M. tb* cultures grown in media (in vitro) [39,40]. Consequently, Rv0976c is considered one of the in vivo gene signatures of *M. tb* and likely plays a role in host infection [39,40]. Moreover, our unpublished data revealed that the expression levels of Rv0976c and several other genes of the iVEGI in *M. tb* grown in vitro were significantly increased when the function of Lsr2 was abolished by target deletion. Lsr2 is a nucleoid-associated protein that preferentially binds DNAs with low GC-content in the *M. tb* genome and silences their expression [23,24]. The GC content of the iVEGI is <55%, which is significantly lower than the average GC-content (65%) of the *M. tb* genome [39]. Thus, it is not surprising that Lsr2 controls the expression of genes in the iVEGI, including Rv0976c. Lsr2 was recently shown to play a role in *M. tb* persistent infection. These pieces of evidence suggest that Rv0976c is highly expressed in *M. tb* during host infection and likely plays a role in latent infection, making it an attractive candidate for testing vaccines. 

## 5. Conclusions

Indeed, our experimental results showed that Rv0976c was highly immunogenic and provided a significant level of protection against *M. tb* challenge. Taken together, these results indicate that Rv0976c is a protective antigen that could be included in novel subunit TB vaccines. Future studies combining Rv0976c with other antigens and evaluating its effectiveness as a booster of BCG or as a therapeutic vaccine will determine its full potential.

## Figures and Tables

**Figure 1 vaccines-13-00872-f001:**
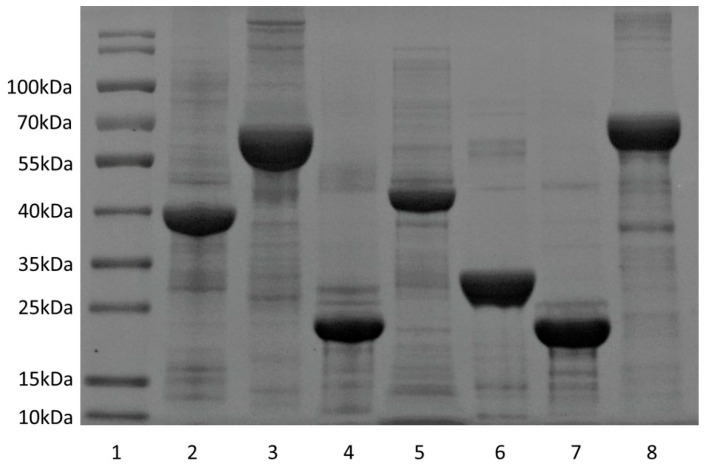
Examples of purified proteins. Lane 1: molecular weight standard; lane 2: Ag85A; lane 3: Rv0976c; lane 4: Rv1506c; lane 5: PPE18; lane 6: Rv1505c; lane 7: Rv2035; lane 8: FadD21.

**Figure 2 vaccines-13-00872-f002:**
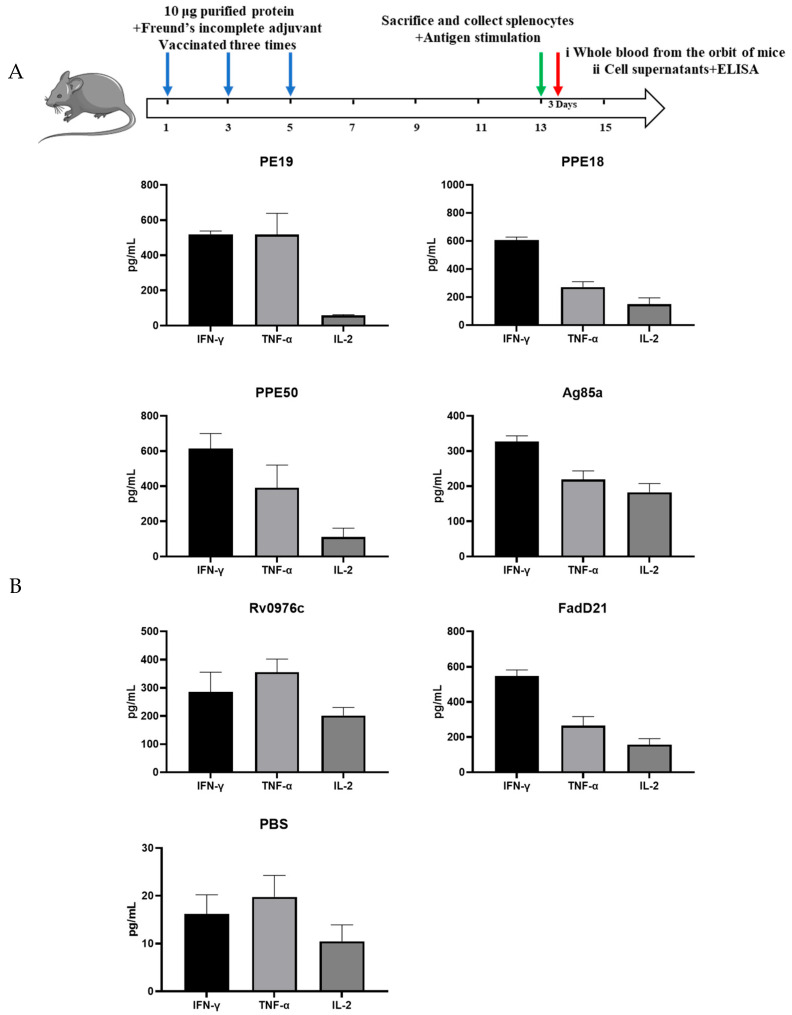
Th1 cytokines (IFN-γ, TNF, IL-2) induced by selected antigens. (**A**) Graphic mouse immunization procedure. (**B**) Mice were immunized three times (2 weeks apart) with 10 μg purified protein formulated with Freund’s incomplete adjuvant and the productions of Th1 cytokines in splenocytes were determined. Data were plotted as mean ± SEM (*n* = 4).

**Figure 3 vaccines-13-00872-f003:**
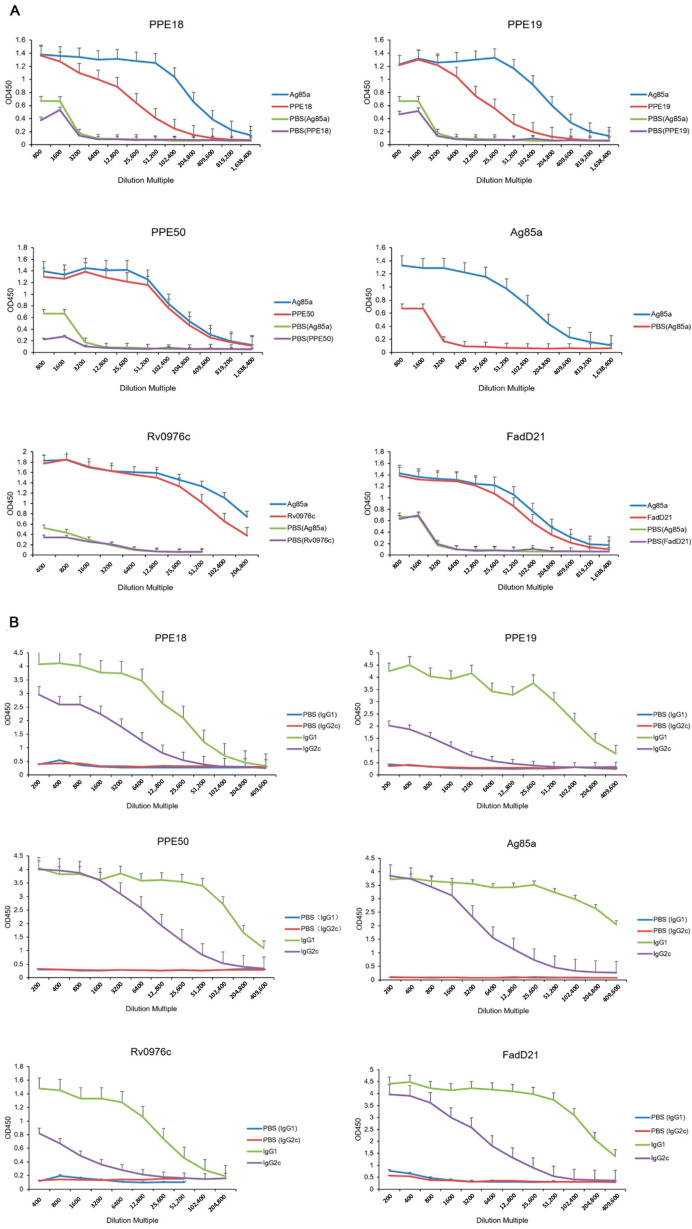
IgG (IgG1 and IgG2c) induced by selected antigens. Mice were immunized subcutaneously three times (2 weeks apart) with 10 μg purified protein formulated with Freund’s incomplete adjuvant and the productions of total IgG (**A**), IgG1, and IgG2c (**B**) in serum were determined. Data were plotted as mean ± SEM (*n* = 4).

**Figure 4 vaccines-13-00872-f004:**
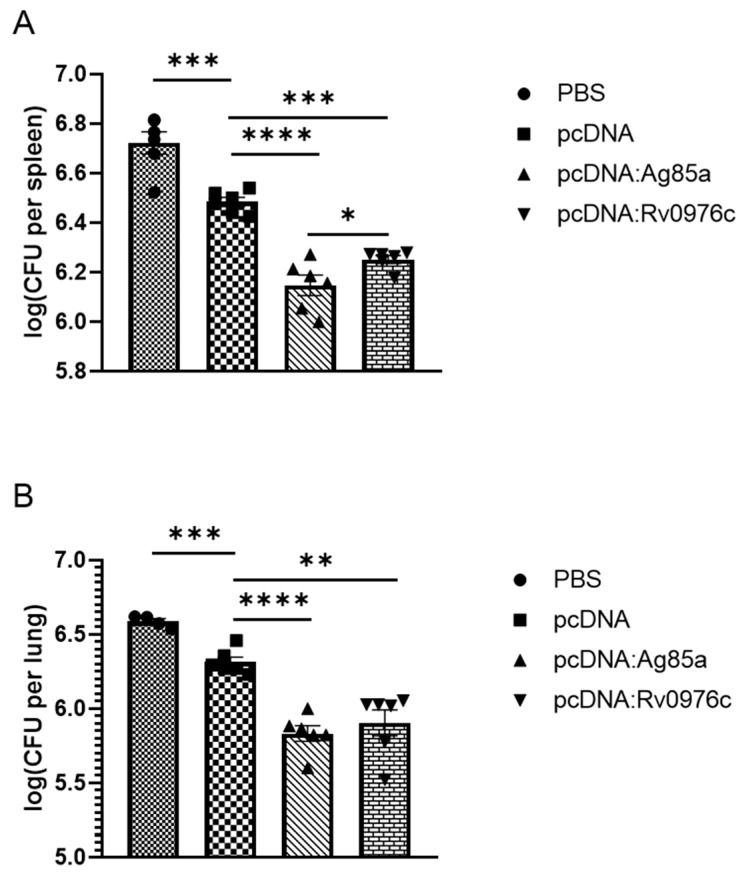
Protective efficacy of selected antigens in BCG Pasteur intravenous infection model. BALB/c mice were immunized with DNA constructs followed by intravenous challenging with 1.47 × 10^7^ CFUs of BCG Pasteur. The *bacteria* burden in the lungs and spleens of each group of animals at week 3 post-infection was determined. Data were plotted as mean ± SEM (*n* = 5). *, *p* < 0.05, **, *p* < 0.01, ***, *p* < 0.001, ****, *p* < 0.0001, unpaired Student’s *t*-test.

**Figure 5 vaccines-13-00872-f005:**
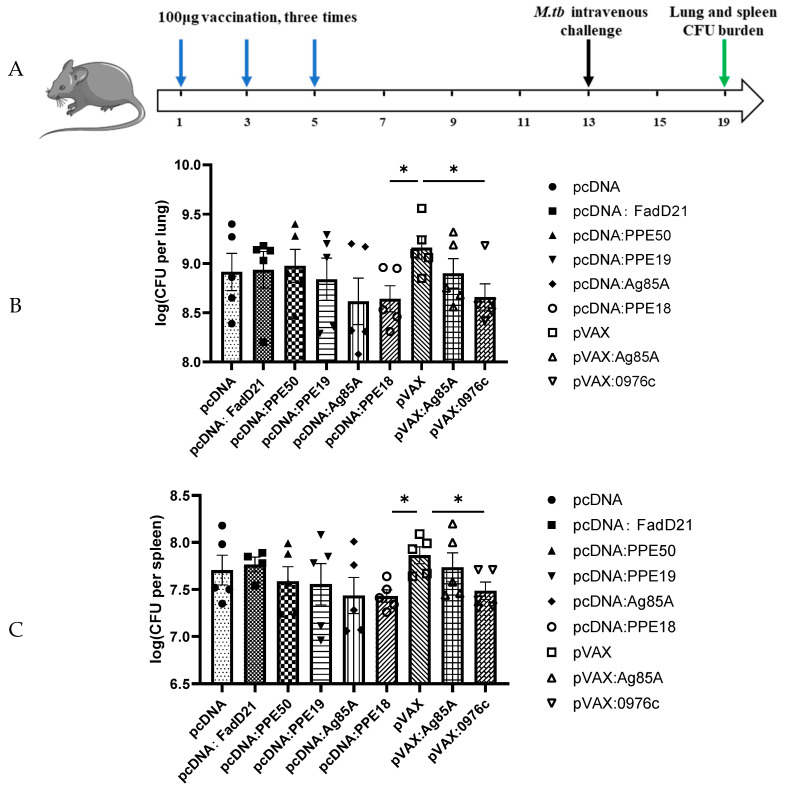
Protective efficacy of selected antigens in BCG Pasteur intravenous infection model. (**A**) Graphic mouse immunization procedure. (**B**,**C**) BALB/c mice were immunized with DNA constructs followed by intravenous challenging with 10^5^ CFUs of *M. tb* H37Rv. The *M. tb* burden in the lungs and spleen of each group of animals at week 4 post-infection was determined. Data were plotted as mean ± SEM (*n* = 5). *, *p* < 0.05, unpaired Student’s *t*-test.

## Data Availability

The original contributions presented in this study are included in the article. Further inquiries can be directed to the corresponding author.

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
