# Peer review of "Protective Efficacy of Subunit Vaccine Expressing Rv0976c Against Tuberculosis"

_vaccines, 2025, doi:10.3390/vaccines13080872_

Round 1
Reviewer 1 Report
Comments and Suggestions for Authors
Comments to the authors:
Zhou et al studied the vaccine potential of Rv1506c, Rv1505c, Rv0976c, and Rv2035; PPE19 and PPE50; and FadD21 using recombinant protein or subunit vaccine. The study further narrowed down on Rv0976c and showed its protective role against Mtb infection. Rv0976c reduces Mtb load and elicits IFN-gamma response, and therefore, authors suggested that Rv0976c may serve as a vaccine candidate. However, I invite authors to address following concerns to improve the manuscript-
Major comments
- Abstract is confusing based on the study design and results- rather including all antigens focus on one Rv0976c and other antigens for supporting the result or find a way to reshape the abstract that matches with major finding of this article.
- Authors choose 7 different antigens to study their protective roles against Mtb infection, however, they have been poorly described in the introduction section.
- The rationale of the study should be more visible in the introduction.
- Solenocytes were isolated from immunized mice followed by again in vitro treatment with the antigens- what is the rationale of this strategy should be described.
- Graphical presentation of experiment strategy should be provided with mice treatment and days when samples were collected.
- From the patients why author tested only Rv0976c not other antigens considering all were tested as immune booster against Mtb?
- Does author test that purified proteins are endotoxin free before administration into mice.
- Why Lsr2 deletion mutant experiment is required here- just for the sake that they are regulators of Rv1506c, Rv1505c, Rv0976c, and Rv2035. Justify this experiment is essential for this manuscript.
- In method part add about reagents procurement, which tubes were used to collect blood to purify serum etc.
- Authors emphasize the protective role of IFN-gamma but they didn’t measure its level in their experiments directly from tissue samples? How this experiment will differ from Fig. 2.
- What are cytokine levels in the vaccine-draining lymph node?
- The expression of antigens from DNA construct should be provided before Mtb infection of mice.
- What about if authors co-administered antigens and BCG role in synergistic protection.
Minor comments-
Writing and English should be revised.
Comments on the Quality of English LanguagePoor
Author Response
Please see the attachment,thank you for your sincere help.

Reviewer 2 Report
Comments and Suggestions for Authors
This study is well-formulated to test the protective efficacy of a subunit vaccine expressing Rv0976c, providing an alternative vaccine against tuberculosis. I appreciate the author's effort in carrying out such a relevant and timely study and preparing the present manuscript. Overall, the study results are convincing in support of Rv0976c as a better alternative subunit vaccine against tuberculosis. However, future studies combining Rv0976c with other antigens and evaluating its effectiveness as a booster of BCG or as a therapeutic vaccine are warranted.
The present manuscript is well-written, and the following are specific comments to improve the manuscript further.
- Line no 52: INF-γ to IFN-γ.
- Consider providing the catalog numbers of the reagents used in the present study. This may be helpful for researchers who want to conduct similar studies in the future.
- How the doses of immunization using purified proteins were determined. Modifying the dose input of this purified protein may have an additional protective efficacy. Consider providing a discussion on this, or if any titration efficacy tests have been conducted, they may be included in the supplementary portion.
- Figure 3 warrants resolution improvement.
- Consider providing information about the route of DNA vulcanization in line 243; however, it appears as subcutaneous in line 264. Also, provide information on the selection criteria for 100 μg of DNA for vaccination. Were all three doses 100ug each, or was there a variation? Is it possible to obtain variable protective efficacy by modulating the doses or booster doses of DNA vaccination?
- Consider also improving the quality of Figure 5.
Author Response
Please see the attachment,thank you very much for your sincere help.

Reviewer 3 Report
Comments and Suggestions for Authors
The manuscript is interesting; it proposes new options for improving immunity against M. tuberculosis infection. The work is well designed and structured. Only a few misspelled words should be corrected, especially the scientific names of the microorganisms, which should be in italics, the genus should be capitalized, and Latin words such as "in vitro" should also be italicized.
Author Response
Thank you very much for your careful reading and patient suggestions. We have corrected the formatting issues.
